# Convergent Block Coordinate Descent for Training Tikhonov Regularized Deep Neural Networks

**Ziming Zhang and Matthew Brand**
Mitsubishi Electric Research Laboratories (MERL)
Cambridge, MA 02139-1955
{zzhang, brand}@merl.com

## Abstract

By lifting the ReLU function into a higher dimensional space, we develop a smooth multi-convex formulation for training feed-forward deep neural networks (DNNs). This allows us to develop a block coordinate descent (BCD) training algorithm consisting of a sequence of numerically well-behaved convex optimizations. Using ideas from proximal point methods in convex analysis, we prove that this BCD algorithm will converge globally to a stationary point with R-linear convergence rate of order one. In experiments with the MNIST database, DNNs trained with this BCD algorithm consistently yielded better test-set error rates than identical DNN architectures trained via all the stochastic gradient descent (SGD) variants in the Caffe toolbox.

## 1 Introduction

Feed-forward deep neural networks (DNNs) are function approximators wherein weighted combinations inputs are filtered through nonlinear activation functions that are organized into a cascade of fully connected (FC) hidden layers. In recent years DNNs have become the tool of choice for many research areas such as machine translation and computer vision.

The objective function for training a DNN is highly non-convex, leading to numerous obstacles to global optimization [10], notably proliferation of saddle points [11] and prevalence of local extrema that offer poor generalization off the training sample [8]. These observations have motivated regularization schemes to smooth or simplify the energy surface, either explicitly such as weight decay [23] or implicitly such as dropout [32] and batch normalization [19], so that the solutions are more robust, *i.e.* better generalized to test data.

Training algorithms face many numerically difficulties that can make it difficult to even find a local optimum. One of the well-known issues is so-called vanishing gradient in back propagation (chain rule differentiation) [18], *i.e.* the long dependency chains between hidden layers (and corresponding variables) tend to drive gradients to zero far from the optimum. This issue leads to very slow improvements of the model parameters, an issue that becomes more and more serious in deeper networks [16]. The vanishing gradient problem can be partially ameliorated by using non-saturating activation functions such as rectified linear unit (ReLU) [25], and network architectures that have shorter input-to-output paths such as ResNet [17]. The saddle-point problem has been addressed by switching from deterministic gradient descent to stochastic gradient descent (SGD), which can achieve weak convergence in probability [6]. Classic proximal-point optimization methods such as the alternating direction method of multipliers (ADMM) have also shown promise for DNN training [34; 41], but in the DNN setting their convergence properties remain unknown.

**Contributions:** In this paper,

1. We propose a novel Tikhonov regularized multi-convex formulation for deep learning, which can be used to learn both dense and sparse DNNs;

2. We propose a novel block coordinate descent (BCD) based learning algorithm accordingly, which can guarantee to globally converge to stationary points with R-linear convergence rate of order one;

3. We demonstrate empirically that DNNs estimated with BCD can produce better representations than DNNs estimated with SGD, in the sense of yielding better test-set classification rates.

Our Tikhonov regularization is motivated by the fact that the ReLU activation function is equivalent to solving a smoothly penalized projection problem in a higher-dimensional Euclidean space. We use this to build a Tikhonov regularization matrix which encodes all the information of the networks, *i.e.* the architectures as well as their associated weights. In this way our training objective can be divided into three sub-problems, namely, (1) Tikhonov regularized inverse problem [37], (2) least-square regression, and (3) learning classifiers. Since each sub-problem is convex and coupled with the other two, our overall objective is multi-convex.

Block coordinate descent (BCD) is often used for problems where finding an exact solution of a sub-problem with respect to a subset (block) of variables is much simpler than finding the solution for all variables simultaneously [27]. In our case, each sub-problem isolates block of variables which can be solved easily (*e.g.* close-form solutions exist). One of the advantages of our decomposition into sub-problems is that the long-range dependency between hidden layers is captured within a sub-problem whose solution helps to propagate the information between inputs and outputs to stabilize the networks (*i.e.* convergence). Therefore, *it does not suffer from vanishing gradient at all.* In our experiments, we demonstrate the effectiveness and efficiency of our algorithm by comparing with SGD based solvers.

## 1.1 Related Work

**(1) Stochastic Regularization (SR) *vs.* Local Regularization *vs.* Tikhonov Regularization:** SR is a widely-used technique in deep learning to prevent the training from overfitting. The basic idea in SR is to multiple the network weights with some random variables so that the learned network is more robust and generalized to test data. Dropout [32] and its variants such like [22] are classic examples of SR. Gal & Ghahramani [14] showed that SR in deep learning can be considered as approximate variational inference in Bayesian neural networks.

Recently Baldassi *et al.* [2] proposed smoothing non-convex functions with local entropy, and latter Chaudhari *et al.* [8] proposed Entropy-SGD for training DNNs. The idea behind such methods is to locate solutions locally within large flat regions of the energy landscape that favors good generalization. In [9] Chaudhari *et al.* provided the mathematical justification for these methods from the perspective of partial differential equations (PDEs)

In contrast, our Tikhonov regularization tends to smooth the non-convex loss *explicitly, globally, and data-dependently*. We deterministically learn the Tikhonov matrix as well as the auxiliary variables in the ill-posed inverse problems. The Tikhonov matrix encodes all the information in the network, and the auxiliary variables represent the ideal outputs of the data from each hidden layer that minimize our objective. Conceptually these variables work similarly as target propagation [4].

**(2) SGD *vs.* BCD:** In [6] Bottou *et al.* proved weak convergence of SGD for non-convex optimization. Ghadimi & Lan [15] showed that SGD can achieve convergence rates that scale as $O\left(t^{-1/2}\right)$ for non-convex loss functions if the stochastic gradient is unbiased with bounded variance, where $t$ denotes the number of iterations.

For non-convex optimization, the BCD based algorithm in [39] was proven to converge globally to stationary points. For parallel computing another BCD based algorithm, namely Parallel Successive Convex Approximation (PSCA), was proposed in [31] and proven to be convergent.

**(3) ADMM *vs.* BCD:** Alternating direction method of multipliers (ADMM) is a proximal-point optimization framework from the 1970s and recently championed by Boyd [7]. It breaks a nearly-separable problem into loosely-coupled smaller problems, some of which can be solved independently and thus in parallel. ADMM offers linear convergence for strictly convex problems, and for certain special non-convex optimization problems, ADMM can also converge [29; 36]. Unfortunately, thus

far there is no evidence or mathematical argument that DNN training is one of these special cases. Therefore, even though empirically it has been successfully applied to DNN training [34; 41], it still lacks of convergence guarantee.

Our BCD-based DNN training algorithm is also amenable to ADMM-like parallelization. More importantly, as we prove in Sec. 4, it will converge globally to stationary points with R-linear convergence.

## 2 Tikhonov Regularization for Deep Learning

### 2.1 Problem Setup

**Key Notations:** We denote $\mathbf{x}_i \in \mathbb{R}^{d_0}$ as the $i$-th training data, $y_i \in \mathcal{Y}$ as its corresponding class label from label set $\mathcal{Y}$, $\mathbf{u}_{i,n} \in \mathbb{R}^{d_n}$ as the output feature for $\mathbf{x}_i$ from the $n$-th ($1 \leq n \leq N$) hidden layer in our network, $\mathbf{W}_{n,m} \in \mathbb{R}^{d_n \times d_m}$ as the weight matrix between the $n$-th and $m$-th hidden layers, $\mathcal{M}_n$ as the input layer index set for the $n$-th hidden layer, $\mathbf{V} \in \mathbb{R}^{d_{N+1} \times d_N}$ as the weight matrix between the last hidden layer and the output layer, $\mathcal{U}, \mathcal{V}, \mathcal{W}$ as nonempty closed convex sets, and $\ell(\cdot, \cdot)$ as a *convex* loss function.

**Network Architectures:** In our networks we only consider ReLU as the activation functions. To provide short paths through the DNN, we allow *multi-input ReLU* units which can take the outputs from multiple previous layers as its inputs.

Fig. 1 illustrates a network architecture that we consider, where the third hidden layers (with ReLU activations), for instance, takes the input data and the outputs from the first and second hidden layers as its inputs. Mathematically, we define our multi-input ReLU function at layer $n$ for data $\mathbf{x}_i$ as:

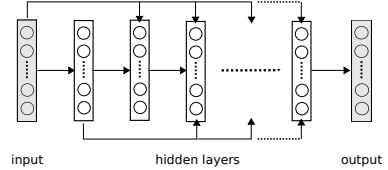
input        hidden layers        output

$$\mathbf{u}_{i,n} = \begin{cases} \mathbf{x}_i, & \text{if } n = 0 \\ \max\left\{\mathbf{0}, \sum_{m \in \mathcal{M}_n} \mathbf{W}_{n,m}\mathbf{u}_{i,m}\right\}, & \text{otherwise} \end{cases} \quad (1)$$

Figure 1: Illustration of DNN architectures that we consider in the paper.

where $\max$ denotes the entry-wise max operator and $\mathbf{0}$ denotes a $d_n$-dim zero vector. Note that multi-input ReLUs can be thought of as conventional ReLU with skip layers [17] where $\mathbf{W}$'s are set to identity matrices accordingly.

**Conventional Objective for Training DNNs with ReLU:** We write down the general objective[1] in a recursive way as used in [41] as follows for clarity:

$$\min_{\mathbf{V} \in \mathcal{V}, \tilde{\mathcal{W}} \subseteq \mathcal{W}} \sum_i \ell(y_i, \mathbf{V}\mathbf{u}_{i,N}), \text{ s.t. } \mathbf{u}_{i,n} = \max\left\{\mathbf{0}, \sum_{m \in \mathcal{M}_n} \mathbf{W}_{n,m}\mathbf{u}_{i,m}\right\}, \mathbf{u}_{i,0} = \mathbf{x}_i, \forall i, \forall n, \quad (2)$$

where $\tilde{\mathcal{W}} = \{\mathbf{W}_{n,m}\}$. Note that we separate the last FC layer (with weight matrix $\mathbf{V}$) from the rest hidden layers (with weight matrices in $\tilde{\mathcal{W}}$) intentionally, because $\mathbf{V}$ is for learning classifiers while $\tilde{\mathcal{W}}$ is for learning useful features. The network architectures we use in this paper are mainly for extracting features, on top of which any arbitrary classifier can be learned further.

Our goal is to optimize Eq. 2. To that end, we propose a novel BCD based algorithm which can solve the relaxation of Eq. 2 using Tikhonov regularization with convergence guarantee.

### 2.2 Reinterpretation of ReLU

The ReLU, ordinarily defined as $\mathbf{u} = \max\{\mathbf{0}, \mathbf{x}\}$ for $\mathbf{x} \in \mathbb{R}^d$, can be viewed as a projection onto a convex set (POCS) [3], and thus rewritten as a simple smooth convex optimization problem,

$$\max\{\mathbf{0}, \mathbf{x}\} \equiv \arg\min_{\mathbf{u} \in \mathcal{U}} \|\mathbf{u} - \mathbf{x}\|_2^2, \quad (3)$$

where $\|\cdot\|_2$ denotes the $\ell_2$ norm of a vector and $\mathcal{U}$ here is the nonnegative closed half-space. This non-negative least squares problem becomes the basis of our lifted objective.

## 2.3 Our Tikhonov Regularized Objective

We use Eq. 3 to lift and unroll the general training objective in Eq. 2 obtaining the relaxation:

$$\min_{\tilde{\mathcal{U}} \subseteq \mathcal{U}, \mathbf{V} \in \mathcal{V}, \tilde{\mathcal{W}} \subseteq \mathcal{W}} f(\tilde{\mathcal{U}}, \mathbf{V}, \tilde{\mathcal{W}}) \triangleq \sum_i \ell(y_i, \mathbf{V}\mathbf{u}_{i,N}) + \sum_{i,n} \frac{\gamma_n}{2} \left\| \mathbf{u}_{i,n} - \sum_{m \in \mathcal{M}_n} \mathbf{W}_{n,m}\mathbf{u}_{i,m} \right\|_2^2, \quad (4)$$
$$\text{s.t.} \qquad \mathbf{u}_{i,n} \geq \mathbf{0}, \mathbf{u}_{i,0} = \mathbf{x}_i, \forall i, \forall n \geq 1,$$

where $\tilde{\mathcal{U}} = \{\mathbf{u}_{i,n}\}$ and $\gamma_n \geq 0, \forall n$ denote predefined regularization constants. Larger $\gamma_n$ values force $\mathbf{u}_{i,n}, \forall i$ to more closely approximate the output of ReLU at the $n$-th hidden layer. Arranging $\mathbf{u}$ and $\gamma$ terms into a matrix $\mathbf{Q}$, we rewrite Eq. 4 in familiar form as a Tikhonov regularized objective:

$$\min_{\tilde{\mathcal{U}} \subseteq \mathcal{U}, \mathbf{V} \in \mathcal{V}, \tilde{\mathcal{W}} \subseteq \mathcal{W}} f(\tilde{\mathcal{U}}, \mathbf{V}, \tilde{\mathcal{W}}) \equiv \sum_i \left\{ \ell(y_i, \mathbf{V}\mathbf{P}\mathbf{u}_i) + \frac{1}{2}\mathbf{u}_i^T \mathbf{Q}(\tilde{\mathcal{W}})\mathbf{u}_i \right\}. \quad (5)$$

Here $\mathbf{u}_i, \forall i$ denotes the concatenating vector of all hidden outputs as well as the input data, *i.e.* $\mathbf{u}_i = [\mathbf{u}_{i,n}]_{n=0}^N, \forall i$, $\mathbf{P}$ is a predefined constant matrix so that $\mathbf{P}\mathbf{u}_i = \mathbf{u}_{i,N}, \forall i$, and $\mathbf{Q}(\tilde{\mathcal{W}})$ denotes another matrix constructed by the weight matrix set $\tilde{\mathcal{W}}$.

**Proposition 1.** $\mathbf{Q}(\tilde{\mathcal{W}})$ *is positive semidefinite, leading to the following Tikhonov regularization:*

$$\mathbf{u}_i^T \mathbf{Q}(\tilde{\mathcal{W}})\mathbf{u}_i \equiv (\mathbf{\Gamma}\mathbf{u}_i)^T(\mathbf{\Gamma}\mathbf{u}_i) = \|\mathbf{\Gamma}\mathbf{u}_i\|_2^2, \exists \mathbf{\Gamma}, \forall i,$$

*where $\mathbf{\Gamma}$ is the Tikhonov matrix.*

**Definition 1** (Block Multi-Convexity [38])**.** *A function $f$ is* block multi-convex *if for each block variable $\mathbf{x}_i, \forall i$, $f$ is a convex function of $\mathbf{x}_i$ while all the other blocks are fixed.*

**Proposition 2.** $f(\tilde{\mathcal{U}}, \mathbf{V}, \tilde{\mathcal{W}})$ *is block multi-convex.*

# 3 Block Coordinate Descent Algorithm

## 3.1 Training

Eq. 4 can be minimized using alternating optimization, which decomposes the problem into the following three convex sub-problems based on Lemma 2:

- Tikhonov regularized inverse problem: $\min_{\mathbf{u}_i \in \mathcal{U}} \ell(y_i, \mathbf{V}\mathbf{P}\mathbf{u}_i) + \frac{1}{2}\mathbf{u}_i^T \mathbf{Q}(\tilde{\mathcal{W}})\mathbf{u}_i, \forall i$.

- Least-square regression: $\min_{\forall \mathbf{W}_{n,m} \in \tilde{\mathcal{W}}} \frac{\gamma_n}{2} \sum_i \left\| \mathbf{u}_{i,n} - \sum_{m \in \mathcal{M}_n} \mathbf{W}_{n,m}\mathbf{u}_{i,m} \right\|_2^2$;

- Classification using learned features: $\min_{\mathbf{V} \in \mathcal{V}} \sum_i \ell(y_i, \mathbf{V}\mathbf{P}\mathbf{u}_i)$.

All the three sub-problems can be solved efficiently due to their convexity. In fact the inverse sub-problem alleviates the vanishing gradient issue in traditional deep learning, because it tries to obtain the *estimated* solution for the output feature of each hidden layer, which are dependent on each other through the Tikhonov matrix. Such functionality is similar to that of target (*i.e.* estimated outputs of each layer) propagation [4], namely, propagating information between input data and output labels.

Unfortunately, a simple alternating optimization scheme cannot guarantee the convergence to stationary points for solving Eq. 4. Therefore we propose a novel BCD based algorithm for training DNNs based on Eq. 4 as listed in Alg. 1. Basically we sequentially solve each sub-problem with an extra quadratic term. These extra terms as well as the convex combination rule guarantee the global convergence of the algorithm (see Sec. 4 for more details).

Our algorithm involves solving a sequence of quadratic programs (QP), whose computational complexity is cubic, in general, in the input dimension [28]. In this paper we focus on the theoretical development of the algorithm, and consider fast implementations in future work.

## 3.2 Testing

Given a test sample $\mathbf{x}$ and learned network weights $\tilde{\mathcal{W}}^*, \mathbf{V}^*$, based on Eq. 4 the ideal decision function for classification should be $y^* = \arg\min_{y \in \mathcal{Y}} \left\{ \min_{\mathbf{u}} f(\mathbf{u}, \mathbf{V}^*, \tilde{\mathcal{W}}^*) \right\}$. This indicates that

**Algorithm 1** Block Coordinate Descent (BCD) Algorithm for Training DNNs

---
**Input** : training data $\{(\mathbf{x}_i, \mathbf{y}_i)\}$ and regularization parameters $\{\gamma_n\}$
**Output** : network weights $\tilde{\mathcal{W}}$

Randomly initialize $\tilde{\mathcal{U}}^{(0)} \subseteq \mathcal{U}, \mathbf{V}^{(0)} \in \mathcal{V}, \tilde{\mathcal{W}}^{(0)} \subseteq \mathcal{W}$;
Set sequence $\{\theta_t\}_{t=1}^\infty$ so that $0 \leq \theta_t \leq 1, \forall t$ and sequence $\left\{ \sum_{k=t}^\infty \frac{\theta_k}{1-\theta_k} \right\}_{t=1}^\infty$ converges to zero, *e.g.* $\theta_t = \frac{1}{t^2}$;
**for** $t = 1, 2, \cdots$ **do**
  $\quad \mathbf{u}_i^* \leftarrow \arg\min_{\mathbf{u}_i \in \mathcal{U}} \ell(y_i, \mathbf{V}^{(t-1)}\mathbf{P}\mathbf{u}_i) + \frac{1}{2}\mathbf{u}_i^T \mathbf{Q}(\tilde{\mathcal{W}}^{(t-1)})\mathbf{u}_i + \frac{1}{2}(1-\theta_t)^2 \|\mathbf{u}_i - \mathbf{u}_i^{(t-1)}\|_2^2, \forall i$;
  $\quad \mathbf{u}_i^{(t)} \leftarrow \mathbf{u}_i^{(t-1)} + \theta_t(\mathbf{u}_i^* - \mathbf{u}_i^{(t-1)}), \forall i$;
  $\quad \mathbf{V}^* \leftarrow \arg\min_{\mathbf{V} \in \mathcal{V}} \sum_i \ell(y_i, \mathbf{V}\mathbf{P}\mathbf{u}_i^{(t)}) + \frac{1}{2}(1-\theta_t)^2 \|\mathbf{V} - \mathbf{V}^{(t-1)}\|_F^2$;
  $\quad \mathbf{V}^{(t)} \leftarrow \mathbf{V}^{(t-1)} + \theta_t(\mathbf{V}^* - \mathbf{V}^{(t-1)})$;
  $\quad \tilde{\mathcal{W}}^* \leftarrow \arg\min_{\tilde{\mathcal{W}} \subseteq \mathcal{W}} \sum_i \frac{1}{2}[\mathbf{u}_i^{(t)}]^T \mathbf{Q}(\tilde{\mathcal{W}})\mathbf{u}_i^{(t)} + \frac{1}{2}(1-\theta_t)^2 \sum_n \sum_{m \in \mathcal{M}_n} \|\mathbf{W}_{n,m} - \mathbf{W}_{n,m}^{(t-1)}\|_F^2$
  $\quad \mathbf{W}_{n,m}^{(t)} \leftarrow \mathbf{W}_{n,m}^{(t-1)} + \theta_t(\mathbf{W}_{n,m}^* - \mathbf{W}_{n,m}^{(t-1)}), \forall n, \forall m \in \mathcal{M}_n, \mathbf{W}_{n,m}^* \in \tilde{\mathcal{W}}^*$;
**end**
**return** $\tilde{\mathcal{W}}$;

---

for each pair of test data and potential label we have to solve an optimization problem, leading to unaffordably high computational complexity that prevents us from using it.

Recall that our goal is to train feed-forward DNNs using the BCD algorithm in Alg. 1. Considering this, we utilize the network weights $\tilde{\mathcal{W}}^*$ to construct the network for extracting deep features. Since these features are the approximation of $\tilde{\mathcal{U}}$ in Eq. 4 (in fact this is a feasible solution of an extreme case where $\gamma_n = +\infty, \forall n$), the learned classifier $\mathbf{V}^*$ can never be reused at test time. Therefore, we retain the architecture and weights of the trained network and replace the classification layer (*i.e.* the last layer with weights $\mathbf{V}$) with a linear support vector machine (SVM).

### 3.3 Experiments

#### 3.3.1 MNIST Demonstration

To demonstrate the effectiveness and efficiency of our BCD based algorithm in Alg. 1, we conduct comprehensive experiments on MNIST [26] dataset using its $28 \times 28 = 784$ raw pixels as input features. We refer to our algorithm for learning *dense* networks as "BCD" and that for learning *sparse* networks as "BCD-S", respectively. For sparse learning, we define the convex set $\mathcal{W} = \{\mathbf{W} \mid \|\mathbf{W}_k\|_1 \leq 1, \forall k\}$, where $\mathbf{W}_k$ denotes the $k$-th row in matrix $\mathbf{W}$ and $\|\cdot\|_1$ denotes the $\ell_1$ norm of a vector. All the comparisons are performed on the same PC. We implement our algorithms using MATLAB GPU implementation without optimizing the code.

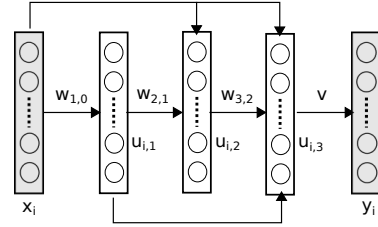

Figure 2: The network architecture for algorithm/solver comparison.

We compare our algorithms with the six SGD based solvers in Caffe [20], *i.e.* SGD [5], AdaDelta [40], AdaGrad [12], Adam [21], Nesterov [33], RMSProp [35], which are coded in Python. The network architecture that we implemented is illustrated in Fig. 2. This network has three hidden layers (with ReLU) with 784 nodes per layer, four FC layers, and three skip layers inside. Therefore, the mapping function from input $\mathbf{x}_i$ to output $\mathbf{y}_i$ defined by the network is:

$$f(\mathbf{x}_i) = \mathbf{V}\mathbf{u}_{i,3}, \ \mathbf{u}_{i,3} = \max\{\mathbf{0}, \mathbf{x}_i + \mathbf{u}_{i,1} + \mathbf{W}_{3,2}\mathbf{u}_{i,2}\},$$
$$\mathbf{u}_{i,2} = \max\{\mathbf{0}, \mathbf{x}_i + \mathbf{W}_{2,1}\mathbf{u}_{i,1}\}, \ \mathbf{u}_{i,1} = \max\{\mathbf{0}, \mathbf{W}_{1,0}\mathbf{x}_i\}.$$

For simplicity without loss of generality, we utilize MSE as the loss function, and learn the network parameters using different solvers with the same inputs and random initial weights for each FC layer.

Without fine-tuning the regularization parameters, we simply set $\gamma_n = 0.1, \forall n$ in Eq. 4 for both BCD and BCD-S algorithms. For the Caffe solvers, we modify the demo code in Caffe for MNIST and run the comparison with carefully tuning the parameters to achieve the best performance that we can. We report the results within 100 epochs by averaging three trials, because at this point the training of all the methods seems convergent already. For all competing algorithms, in each epoch the entire

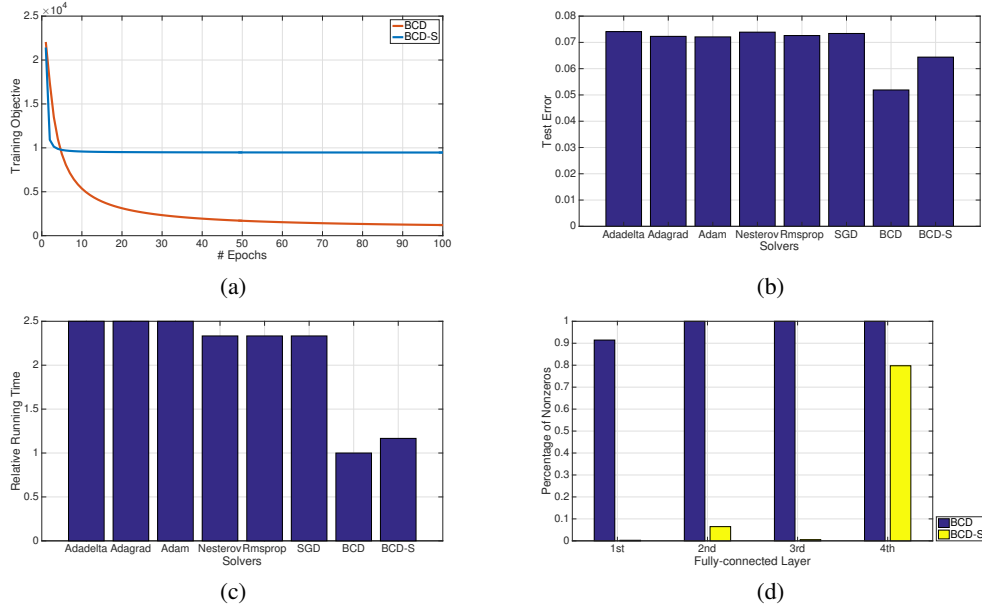

(a)                                              (b)

(c)                                              (d)

Figure 3: **(a)** Illustration of convergence for BCD and BCD-S. **(b)** Test error comparison. **(c)** Running time comparison. **(d)** Sparseness comparison for BCD and BCD-S.

training data is passed through once to update parameters. Therefore, for our algorithms each epoch is equivalent to one iteration, and there are 100 iterations in total.

**Convergence:** Fig. 3(a) shows the change of training objective with increase of epochs for BCD and BCD-S, respectively. As we see both curves decrease monotonically and become flatter and flatter eventually, indicating that both algorithms converge. BCD-S converges much faster than BCD, but its objective is higher than BCD. This is because BCD-S learns sparse models that may not fit data as well as dense models learned by BCD.

**Testing Error:** As mentioned in Sec. 3.2, here we utilize linear SVMs and last-layer hidden features extracted from training data to retrain the classifier. Based on the network in Fig. 2 the feature extraction function is $\mathbf{u}_{i,3} = \max\{\mathbf{0}, \mathbf{x}_i + \max\{\mathbf{0}, \mathbf{W}_{1,0}\mathbf{x}_i\} + \mathbf{W}_{3,2}\max\{\mathbf{0}, \mathbf{x}_i + \mathbf{W}_{2,1}\max\{\mathbf{0}, \mathbf{W}_{1,0}\mathbf{x}_i\}\}\}$. To conduct fair comparison, we retrain the classifiers for all the algorithms, and summarize the test-time results in Fig. 3(b) with 100 epochs. Our BCD algorithm which learns dense architectures, same as the SGD based solvers, performs best, while our BCD-S algorithm works still better than the SGD competitors, although it learns much sparser networks. These results are consistent with the training objectives in Fig. 3(a) as well.

**Computational Time:** We compare the training time in Fig. 3(c). It seems that our BCD implementation is significantly faster than the Caffe solvers. For instance, our BCD achieves about 2.5 times speed-up than the competitors, while achieving best classification performance at test time.

**Sparseness:** In order to compare the difference in terms of weights between the dense and sparse networks learned by BCD and BCD-S, respectively, we compare the percentage of nonzero weights in each FC layer, and show the results in Fig. 3(d). As we see, expect the last FC layer (corresponding to parameter $\mathbf{V}$ as classifiers) BCD-S has the ability of learning much sparser networks for deep feature extraction. In our case BCD-S learns a network with $2.42\%$ nonzero weights[2], on average, with classification accuracy $1.34\%$ lower than that of BCD which learns a network with $97.15\%$ nonzero weights. Potentially this ability could be very useful in the scenarios such as embedding systems where sparse networks are desired.

### 3.3.2 Supervised Hashing

To further demonstrate the usage of our approach, we compare with [41][3] for the application of supervised hashing, which is the state-of-the-art in the literature. [41] proposed an ADMM based

optimization algorithm to train DNNs with relaxed objective that is very related to ours. We train the same DNN on MNIST as used in [41], *i.e.* with 48 hidden layers and 256 nodes per layer that are sequentially and fully connected (see [41] for more details on the network). Using the same image features, we consistently observe marginal improvement over the results (*i.e.* precision, recall, mAP) reported in [41]. However, on the same PC we can finish training within 1 hour based on our implementation, while using the MATLAB code for [41] the training needs about 9 hours. Similar observations can be made on CIFAR-10 as used in [41] using a network with 16 hidden layers and 1024 nodes per layer.

# 4 Convergence Analysis

## 4.1 Preliminaries

**Definition 2** (Lipschitz Continuity [13]). *We say that function $f$ is* Lipschitz continuous *with Lipschitz constant $L_f$ on $\mathcal{X}$, if there is a (necessarily nonnegative) constant $L_f$ such that*

$$|f(x_1) - f(x_2)| \leq L_f |x_1 - x_2|, \forall x_1, x_2 \in \mathcal{X}.$$

**Definition 3** (Global Convergence [24]). *Let $\mathcal{X}$ be a set and $x_0 \in \mathcal{X}$ a given point, Then an Algorithm, $\mathcal{A}$, with initial point $x_0$ is a point-to-set map $\mathcal{A}: \mathcal{X} \to \mathcal{P}(\mathcal{X})$ which generates a sequence $\{x_k\}_{k=1}^{\infty}$ via the rule $x_{k+1} \in \mathcal{A}(x_k), k = 0, 1, \cdots$. $\mathcal{A}$ is said to be* global convergent *if for any chosen initial point $x_0$, the sequence $\{x_k\}_{k=0}^{\infty}$ generated by $x_{k+1} \in \mathcal{A}(x_k)$ (or a subsequence) converges to a point for which a necessary condition of optimality holds.*

**Definition 4** (R-linear Convergence Rate [30]). *Let $\{x_k\}$ be a sequence in $\mathbb{R}^n$ that converges to $x^*$. We say that convergence is* R-linear *if there is a sequence of nonnegative scalars $\{v_k\}$ such that $\|x_k - x^*\| \leq v_k, \forall k$, and $\{v_k\}$ converges Q-linearly to zero.*

**Lemma 1** (3-Point Property [1]). *If function $\phi(\mathbf{w})$ is convex and $\hat{\mathbf{w}} = \arg\min_{\mathbf{w} \in \mathbb{R}^d} \phi(\mathbf{w}) + \frac{1}{2}\|\mathbf{w} - \mathbf{w}_0\|_2^2$, then for any $\mathbf{w} \in \mathbb{R}^d$,*

$$\phi(\hat{\mathbf{w}}) + \frac{1}{2}\|\hat{\mathbf{w}} - \mathbf{w}_0\|_2^2 \leq \phi(\mathbf{w}) + \frac{1}{2}\|\mathbf{w} - \mathbf{w}_0\|_2^2 - \frac{1}{2}\|\mathbf{w} - \hat{\mathbf{w}}\|_2^2.$$

## 4.2 Theoretical Results

**Definition 5** (Assumptions on $f$ in Eq. 4). *Let $f_1(\tilde{\mathcal{U}}) \triangleq f(\tilde{\mathcal{U}}, \cdot, \cdot), f_2(\mathbf{V}) \triangleq f(\cdot, \mathbf{V}, \cdot), f_3(\tilde{\mathcal{W}}) \triangleq f(\cdot, \cdot, \tilde{\mathcal{W}})$ be the objectives of the three sub-problems, respectively. Then we assume that $f$ is lower-bounded and $f_1, f_2, f_3$ are Lipschitz continuous with constants $L_{f_1}, L_{f_2}, L_{f_3}$, respectively.*

**Proposition 3.** *Let $x, y, \hat{x} \in \mathcal{X}$ and $y = (1 - \theta)x + \theta\hat{x}$. Then $\frac{1}{2}\|\hat{x} - y\|_2^2 = \frac{1}{2}(1 - \theta)^2 \|\hat{x} - x\|_2^2$.*

**Lemma 2.** *Let $\mathcal{X}$ be a nonempty closed convex set, function $\phi: \mathcal{X} \to \mathbb{R}$ is convex and Lipschitz continuous with constant $L$, and scalar $0 \leq \theta \leq 1$. Suppose that $\forall x \in \mathcal{X}, \hat{x} = \arg\min_{z \in \mathcal{X}} \phi(z) + \frac{1}{2}\|z - z_0\|_2^2$ and $z_0 = y = (1 - \theta)x + \theta\hat{x}$. Then we have*

$$\frac{1 - \theta}{\theta}\|y - x\|_2^2 \leq \phi(x) - \phi(y) \leq L\|y - x\|_2 \Rightarrow \|y - x\|_2 \leq \frac{L\theta}{1 - \theta}.$$

*Proof.* Based on the convexity of $\phi$, Prop. 3, and Lemma 1, we have

$$\phi(x) - \phi(y) \geq \phi(x) - [(1 - \theta)\phi(x) + \theta\phi(\hat{x})] = \theta[\phi(x) - \phi(\hat{x})]$$

$$\geq \theta\left[\frac{1}{2}\|x - \hat{x}\|_2^2 + \frac{1}{2}\|\hat{x} - z_0\|_2^2 - \frac{1}{2}\|x - z_0\|_2^2\right] = \theta(1 - \theta)\|x - \hat{x}\|_2^2 = \frac{1 - \theta}{\theta}\|y - x\|_2^2,$$

where $\|y - x\|_2^2 = 0$ if and only if $\hat{x} = x$ (equivalently $\phi(x) = \phi(y)$); otherwise $\|y - x\|_2^2$ is lower-bounded from 0 provided that $\theta \neq 1$.

Based on Def. 2, we have $\phi(x) - \phi(y) \leq L\|y - x\|_2$. $\square$

**Theorem 1.** *Let $\left\{\left(\tilde{\mathcal{U}}^{(t)}, \mathbf{V}^{(t)}, \tilde{\mathcal{W}}^{(t)}\right)\right\}_{t=1}^{\infty} \subseteq \mathcal{U} \times \mathcal{V} \times \mathcal{W}$ be an arbitrary sequence from a closed convex set that is generated by Alg. 1. Suppose that $0 \leq \theta_t \leq 1, \forall t$ and the sequence $\left\{\sum_{k=t}^{\infty} \frac{\theta_k}{1 - \theta_k}\right\}_{t=1}^{\infty}$ converges to zero. Then we have*

1. $\left(\tilde{\mathcal{U}}^{(\infty)}, \mathbf{V}^{(\infty)}, \tilde{\mathcal{W}}^{(\infty)}\right)$ is a stationary point;

2. $\left\{\left(\tilde{\mathcal{U}}^{(t)}, \mathbf{V}^{(t)}, \tilde{\mathcal{W}}^{(t)}\right)\right\}_{t=1}^{\infty}$ will converge to $\left(\tilde{\mathcal{U}}^{(\infty)}, \mathbf{V}^{(\infty)}, \tilde{\mathcal{W}}^{(\infty)}\right)$ globally with R-linear convergence rate.

*Proof.* 1. Suppose that for $\tilde{\mathcal{U}}^{(\infty)}$ there exists a $\triangle\tilde{\mathcal{U}} \neq \emptyset$ so that $f_1(\tilde{\mathcal{U}}^{(\infty)} + \triangle\tilde{\mathcal{U}}) = f_1(\tilde{\mathcal{U}}^{(\infty)})$ (otherwise, it conflicts with the fact of $\tilde{\mathcal{U}}^{(\infty)}$ being the limit point). From Lemma 2, $f_1(\tilde{\mathcal{U}}^{(\infty)} + \triangle\tilde{\mathcal{U}}) = f_1(\tilde{\mathcal{U}}^{(\infty)})$ is equivalent to $\tilde{\mathcal{U}}^{(\infty)} + \triangle\tilde{\mathcal{U}} = \tilde{\mathcal{U}}^{(\infty)}$, and thus $\triangle\tilde{\mathcal{U}} = \emptyset$, which conflicts with the assumption of $\triangle\tilde{\mathcal{U}} \neq \emptyset$. Therefore, there is no direction that can decrease $f_1(\tilde{\mathcal{U}}^{(\infty)})$, *i.e.* $\nabla f_1(\tilde{\mathcal{U}}^{(\infty)}) = \mathbf{0}$. Similarly we have $\nabla f_2(\mathbf{V}^{(\infty)}) = \mathbf{0}$ and $\nabla f_3(\tilde{\mathcal{W}}^{(\infty)}) = \mathbf{0}$. Therefore, $\left(\tilde{\mathcal{U}}^{(\infty)}, \mathbf{V}^{(\infty)}, \tilde{\mathcal{W}}^{(\infty)}\right)$ is a stationary point.

2. Based on Def. 5 and Lemma 2, we have

$$\sqrt{\sum_{\mathbf{u}_{i,n}\in\tilde{\mathcal{U}}} \left\|\mathbf{u}_{i,n}^{(t)} - \mathbf{u}_{i,n}^{(\infty)}\right\|_2^2 + \left\|\mathbf{V}^{(t)} - \mathbf{V}^{(\infty)}\right\|_F^2 + \sum_{\mathbf{W}_{n,m}\in\tilde{\mathcal{W}}} \left\|\mathbf{W}_{n,m}^{(t)} - \mathbf{W}_{n,m}^{(\infty)}\right\|_F^2}$$

$$\leq \sum_{\mathbf{u}_{i,n}\in\tilde{\mathcal{U}}} \left\|\mathbf{u}_{i,n}^{(t)} - \mathbf{u}_{i,n}^{(\infty)}\right\|_2 + \left\|\mathbf{V}^{(t)} - \mathbf{V}^{(\infty)}\right\|_F + \sum_{\mathbf{W}_{n,m}\in\tilde{\mathcal{W}}} \left\|\mathbf{W}_{n,m}^{(t)} - \mathbf{W}_{n,m}^{(\infty)}\right\|_F$$

$$= \sum_{\mathbf{u}_{i,n}\in\tilde{\mathcal{U}}} \left\|\sum_{k=t}^{\infty} \mathbf{u}_{i,n}^{(k)} - \mathbf{u}_{i,n}^{(k+1)}\right\|_2 + \left\|\sum_{k=t}^{\infty} \mathbf{V}^{(k)} - \mathbf{V}^{(k+1)}\right\|_F + \sum_{\mathbf{W}_{n,m}\in\tilde{\mathcal{W}}} \left\|\sum_{k=t}^{\infty} \mathbf{W}_{n,m}^{(k)} - \mathbf{W}_{n,m}^{(k+1)}\right\|_F$$

$$\leq \sum_{k=t}^{\infty} \left[\sum_{\mathbf{u}_{i,n}\in\tilde{\mathcal{U}}} \left\|\mathbf{u}_{i,n}^{(k)} - \mathbf{u}_{i,n}^{(k+1)}\right\|_2 + \left\|\mathbf{V}^{(k)} - \mathbf{V}^{(k+1)}\right\|_F + \sum_{\mathbf{W}_{n,m}\in\tilde{\mathcal{W}}} \left\|\mathbf{W}_{n,m}^{(k)} - \mathbf{W}_{n,m}^{(k+1)}\right\|_F\right]$$

$$\leq \sum_{k=t}^{\infty} \left[\sum_{\mathbf{u}_{i,n}\in\tilde{\mathcal{U}}} \frac{L_{f_1}\theta_k}{1-\theta_k} + \frac{L_{f_2}\theta_k}{1-\theta_k} + \sum_{\mathbf{W}_{n,m}\in\tilde{\mathcal{W}}} \frac{L_{f_3}\theta_k}{1-\theta_k}\right] = O\left(\sum_{k=t}^{\infty} \frac{\theta_k}{1-\theta_k}\right).$$

By combining this with Def. 3 and Def. 4 we can complete the proof. $\qquad\square$

**Corollary 1.** *Let* $\theta_t = \left(\frac{1}{t}\right)^p, \forall t$. *Then when* $p > 1$, *Alg. 1 will converge globally with order one.*

*Proof.*

$$\sum_{k=t}^{\infty} \frac{\theta_k}{1-\theta_k} = \sum_{k=t}^{\infty} \frac{1}{k^p - 1} \leq \int_{t^p-1}^{\infty} \frac{1}{x} d(x+1)^{\frac{1}{p}} = \frac{1}{p} \int_{t^p-1}^{\infty} \frac{1}{x}(x+1)^{\frac{1}{p}-1} dx$$

$$\overset{\because p>1}{\leq} \frac{1}{p} \int_{t^p-1}^{\infty} x^{\frac{1}{p}-2} dx = (p-1)^{-1}(t^p - 1)^{\frac{1}{p}-1}. \tag{6}$$

Since the sequence $\left\{(t^p - 1)^{\frac{1}{p}-1}\right\}_{t=1}^{\infty}, \forall p > 1$ converges to zero sublinearly with order one, by combining these with Def. 4 and Thm. 1 we can complete the proof. $\qquad\square$

## 5 Conclusion

In this paper we first propose a novel Tikhonov regularization for training DNNs with ReLU as the activation functions. The Tikhonov matrix encodes the network architecture as well as parameterization. With its help we reformulate the network training as a block multi-convex minimization problem. Accordingly we further propose a novel block coordinate descent (BCD) based algorithm, which is proven to converge globally to stationary points with R-linear converge rate of order one. Our empirical results suggest that our algorithm does converge, is suitable for learning both dense and sparse networks, and may work better than traditional SGD based deep learning solvers.

## Footnotes

[1]For simplicity in this paper we always presume that the domain of each variable contains the regularization, *e.g.* $\ell_2$-norm, without showing it in the objective explicitly.

[2]Since we will retrain the classifiers after all, here we do not take the nonzeros in the last FC into account.

[3]MATLAB code is available at `https://zimingzhang.wordpress.com/publications/`.

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
