[Reviews · NeurIPS 2017]

Reviewer 1



This paper proposes a simple and efficient block coordinate descent (BCD) algorithm with a novel Tikhonov regularization for training both dense and sparse DNNs with ReLU. They show that the proposed BCD algorithm converges globally to a stationary point with an R-linear convergence rate of order one and performs better than all the SGD variants in experiments. However, the motivations of using Tikhonov regularization and block coordinate descent are not very clear. The technical parts are hard to follow due to the absence of many details. The presented results are far from state-of-the-art. In this sense, I am not sure whether the proposed method can be applied to real "DNNs". My detailed comments are as follows. Specific comments: 1. The motivation of Tikhonov Regularization should be clarified. Why do we need to introduce Tikhonov Regularization instead of others? What is the main advantage of Tikhonov Regularization compared with other Regularizations? In the second paragraph of Section I, the authors mentioned several issues of DNN, such as the "highly non-convex", "saddle points" and "local extrema", followed by the Tikhonov Regularization. Does it mean that the proposed Tikhonov Regularizationcan address all these issues? 2. What is the motivation of the decomposition into three sub-problems? The authors should explain why such decomposition will not suffer from vanishing gradient. 3. From Eqn. (4) to Eqn. (5), it is hard to follow. On line 133, what is the specific formula of matrix Q(\tilde {\mathcal A})? How to confirm matrix Q(\tilde {\mathcal A}) is positive semidefinite? 4. A mathematical definition of Tikhonov regularized inverse problem should be given clearly. 5. There is a quite strong argument: “In fact, the inverse sub-problem resolve the vanishing gradient issue in traditional deep learning, because it tries to obtain the optimal solution for the output feature of each hidden layer, which are dependent on each other through the Tikhonov matrix.” There are no theoretical justifications or empirical backing to verify that such inverse sub-problem can resolve the vanishing gradient issue. 6. The authors should give more details for why the “optimal” output features work similarly as target propagation. 7. On line 246, what is the definition of \mathcal P? 8. In Fig. 5, BCD-S can learn much sparser networks, why weight matrix of the 4th layer is still dense. 9. On line 252, what the prefix “Q” means? 10. On line 284, there is a missing word: “dense”. 11. The format of the reference is confusing. 12. It is unclear why using the DNN structure in Figure 1. The network studied in the experiments has only 3 hidden layers, which is actually not "deep". Should we use skip connections here? Moreover, the presented results are far from state-of-the-art. In this sense, I am not sure whether the proposed method can be applied to real "DNNs".

Reviewer 2



The paper pulls together a few important recent ideas on optimizing deep neural networks (as an alternative to popular SGD variants). The derivation of the Tikhonov regularized problem in Eq. (4) and (5) from the recursive objective in Eq. (2) through relaxing the ReLU outputs as a convex projection is very clean and convincing. The decomposition into an inverse problem (activations), a least-squares problem (weights) and a final classification problem (soft-max weights) is illuminating. This formulation provides a novel perspectives on deep learning and offers many interesting avenues, e.g. generalization of ReLUs, more general connectivity patterns through shortcuts (as suggested in the used architecture), sparsity in network connectivity, etc. When it comes to optimization, the paper points out that it is difficult to guarantee convergence of a naive alternating optimization scheme (e.g. ADMM) and thus resort to an easier to analyze block-coordinate descent algorithm, presented in Algorithm 1 and analyzed in Section 4. On the plus side: the authors are able to derive a convergent algorithm. On the minus side: it seems that there is a lot more to explore here and that the paper is merely a first (somewhat preliminary) step towards exploiting the derived problem formulation. A proof of concept on MNIST data is given in Section 3. One has to say, that the experimental section is relatively weak: one data set, matlab vs. phython code, very limited analysis. This requires a more extensive quantitative (run-time, solution quality) and qualitativ investigation. Figure 4 - a percentage pie chart is really confusing. For the final version, a more extensive evaluation would greatly improve the quality of the paper.

Reviewer 3



This manuscript proposes a modification of feed-forward neural network optimization problem when the activation function is the ReLU function. My major concern is about the convergence proof for the block coordinate descent algorithm. In particular, Theorem 1 is incorrect for the following reason: A. It assumed that the sequence generated by the algorithm has a limit point. This may not be the case as the set U is not compact (closed but not bounded). Therefore, the sequence may not converge to a finite point. B. It stated that the sequence generated by the algorithm has a unique limit point (line 263: "the" limit point). Even the algorithm has limit points, it may not be unique. Consider the sequence x_i = (-1)^n - 1/n, clearly it has 2 limit points: +1 and -1. For the rest parts of the paper, my comments are as follows. 1. The new formulation seems interesting, and it can be separately discussed from the block coordinate descent part, and it looks to me the major novelty is the new formulation but not the algorithm. However, the major concern for the new formulation is that it has much more variables, which can potentially lead to spatial unfeasibility when there are more data instances. This problem hinders people from applying the formulation to large-scale problems and thus its usefulness is limited. 2. I do not see that the problem in line 149 is always convex with respect to W. This requires further elaboration. 3. In the introduction, some efforts were put in describing the issue of saddle points. However, this manuscript does not handle this problem as well. I'd suggest the authors to remove the related discussion. 4. The format of paper reference and equation reference is rather non-standard. Usually [1] is used for paper reference and (1) is for equation reference. 5. The comparison with Caffe solvers in term of time is meaningless as the implementation platforms are totally different. I do not understand the sentence "using MATLAB still run significantly faster" especially regarding the word "still", as matlab should be expected to be faster than python. 6. The comparison of the objective value in 3(a) is comparing apples to oranges, as the two are solving different problems. === After feedback === I did notice that in Line 100 the authors mentioned that U,V,W are compact sets, but as we see in line 124, U is set to be the nonnegative half-space, which is clearly unbounded and thus non-compact. Therefore, indeed when U is compact, this is correct that there will be at least a limit point and thus the convergence analysis may hold true (see details below), it is not the case for the problem (4) or (5). Regarding the proof, I don't agree with the concept of "the" limit point when t = \infty. I don't think such thing exists by nature. As stated in my original review, there might be several limit points, and by definition we always have all limit points are associated with t = \infty, just in different ways. The authors said in the rebuttal that because their algorithm converges so "the" limit point exists, but this is using the result proved under this assumption to prove that the assumption is true. I don't think that is a valid way of proof. In addition, in the extremest case, consider that the sets U,V,W all consist of one point, say u,v,w, and the gradient at (u,v,w) is not zero. Clearly in this case there is one and only one limit point (u,v,w), but apparently it is not a stationary point as the gradient is non-zero. Therefore, I will keep the same score.